# Information decay and enzymatic information recovery for DNA data storage

Linda C. Meiser[1], Andreas L. Gimpel [1], Tejas Deshpande[1], Gabriela Libort[1], Weida D. Chen[1], Reinhard Heckel[2], Bichlien H. Nguyen[3], Karin Strauss [3], Wendelin J. Stark[1] & Robert N. Grass [1]✉

Synthetic DNA has been proposed as a storage medium for digital information due to its high theoretical storage density and anticipated long storage horizons. However, under all ambient storage conditions, DNA undergoes a slow chemical decay process resulting in nicked (broken) DNA strands, and the information stored in these strands is no longer readable. In this work we design an enzymatic repair procedure, which is applicable to the DNA pool prior to readout and can partially reverse the damage. Through a chemical understanding of the decay process, an overhang at the 3' end of the damaged site is identified as obstructive to repair via the base excision-repair (BER) mechanism. The obstruction can be removed via the enzyme apurinic/apyrimidinic endonuclease I (APE1), thereby enabling repair of hydrolytically damaged DNA via Bst polymerase and Taq ligase. Simulations of damage and repair reveal the benefit of the enzymatic repair step for DNA data storage, especially when data is stored in DNA at high storage densities (=low physical redundancy) and for long time durations.

[1] Institute for Chemical and Bioengineering, Department of Chemistry and Applied Biosciences, ETH Zurich, Vladimir-Prelog-Weg 1, CH-8093 Zurich, Switzerland. [2] Department of Electrical and Computer Engineering, Technical University of Munich, Arcisstrasse 21, 80333 Munich, Germany. [3] Microsoft Research, Redmond, WA 98052, USA. ✉email: rograss@ethz.ch

Genomic DNA, which is present in all living organisms, can be damaged by extrinsic and intrinsic agents, including factors such as exposure to sunlight, oxidation or hydrolysis and lesions in the DNA strands[1,2]. Although natural proofreading mechanisms are in place, some of these lesions and errors may remain unattended, which, if not repaired, can be mutagenic[3].

Three basic cellular defense mechanisms, namely direct repair, base-excision repair (BER), and nucleotide-excision repair (NER), are known, in which specific enzymes repair spontaneous DNA lesions. These lesions could be caused by sunlight, oxidation, hydrolysis or exposure to small molecules. In the absence of large quantities of external DNA-damaging agents, most DNA lesions are repaired by the base-excision-repair pathway, which has been reconstituted by Dianov and Lindahl in 1994[2], using purified enzymes, for which Lindahl jointly received the Nobel Prize in Chemistry in 2015[4].

Since the identification of enzymes involved in the base-excision-repair pathway[2,3,5–7], repair enzymes restoring the quality of degraded DNA have not only been used for understanding genomic DNA repair but are used in standard molecular biology routines and have also been proposed for improving the analysis of ancient DNA, the genotyping of forensic samples as well as the tracing of foods, just to mention a few[8–11]. One application of enzymatic repair that has not yet received any attention, however, is the repair of synthetic DNA in DNA data-storage applications.

For DNA data storage, digital files (bits) are translated into nucleotides (nt), which can then be synthesized and stored for thousands of years[12–17]. Due to constraints in synthesizing and reading very long DNA strands, data are usually stored in a pool of many relatively short (ca. 150 nt) oligos[18]. Aside of encoding for actual data, every oligo also comprises an index for information organization and amplification primers, which enable random access[15,17] and the handling of the oligos for readout (sequencing preparation). The technology usually uses error-correction codes, which calculate and append redundant information to the original data, so that read/write and storage errors can be compensated for[15,16,18]. Such error-correcting codes may thereby enable the correction of individual base errors (e.g., mutations) within a sequence (inner code), as well as the loss of complete sequences (outer code). While such error-correcting codes enable the perfect recovery of the stored data, they come at the cost of having to write (=synthesize) more nucleotides and oligos than would be necessary in the absence of errors. This does not mean that the DNA pools have to be synthesized at a larger scale (i.e., having more copies per oligo), but that more unique oligos have to be synthesized to enable error correction (e.g., larger outer code redundancy)[15,18,19]. The cost of this scales directly with the number of expected errors[20]. DNA synthesis is currently the bottleneck of the more widespread adoption of DNA data storage[21,22], and consequently synthesis efforts should be minimized. Our specific interest in terms of minimizing errors and, therefore, necessary data redundancy lies in the actual DNA storage process itself.

In this work, we analyze the decay process of synthetic DNA, which is expected to occur during a DNA data-storage scenario on a molecular level. We identify DNA nicking as the process resulting in the greatest data loss and propose enzymatic repair for a potential solution to reverse this data loss. This solution presents a process for information recovery for synthetic DNA in data-storage applications.

During storage, DNA may be exposed to several stresses including UV irradiation, oxidation, hydrolysis, alkylation, ionizing radiation, or mechanical shear[23], but independent of the actual storage format, hydrolysis has been identified as the major

decay reaction[16,24]. This is due to the near omnipresence of atmospheric humidity, and the relatively high reaction rates of DNA with water as opposed to oxygen. In addition, water is the standard medium for all DNA handling steps, as DNA is not soluble in any other solvent than water.

In terms of DNA decay, two decay modes are of special interest: the mutation of individual bases, and the formation of nicks (=strand breaks) of the DNA backbone. While the nature of mutations has been described in the past[20], DNA nicks are especially problematic in the view of information readout, as nicked DNA cannot be amplified via polymerase chain reaction (PCR), a procedure typically employed during random access and DNA readout (sequencing preparation) routines (Fig. 1a). Consequently, any DNA strand comprising at least one single-strand break is not amplified, and is thereby not read. In the context of information integrity, a mutation can result in the loss of up to 2 bits of information, and a nick can result in upwards of 100 bits of data lost (assuming that the sequences are of length more than 50, which they typically are). While these losses can be mitigated by error-correction codes, the cost imbalance of mutations vs. nicks remains: The cost for error-correction codes to correct for mutated bases is relatively low: One substitution error requires at least two symbols of redundancy (inner code, Fig. 1d), and one erasure (missing nucleotide) requires at least one symbol of redundancy one symbol can vary in size, a common choice is 3 nt per symbol)[18,25]. The loss of a whole sequence (as a consequence of a single DNA nick), however, is much more expensive, as the whole information of the sequence is lost. From an error-correction standpoint, this is expensive, as the correction of every nick requires at least one full redundant DNA strand (outer code redundancy, Fig. 1d). Specifically, a nick has an expected cost that is higher by about half of the length of the sequence than the cost of a mutation. In practice, the cost of a nick is therefore typically by a factor of 25–100 higher than that of a mutation. As a result, we were interested in investigating the nicking of DNA during storage.

## Results

We thus investigated the damage of DNA that occurs hydrolytically, by looking at two different sets of DNA, one representing a single oligo (model DNA), and one comprising 7373 oligos, together encoding 115,394 bytes of data (Supplementary Table 1). We naturally aged the DNA in water, at 25 °C and 30 °C for up to 40 days and quantified the amount of amplifiable DNA via qPCR. While various DNA preservation methods would enable slower hydrolysis due to the partial protection of the DNA from water, it may be expected that the major decay-promoting step would still encompass the same hydrolysis reactions, but at a significantly lower rate than in aqueous solution.

To investigate the nicking process, we measured the fragmentation of aged DNA by ssDNA sequencing using Illumina's iSeq100. For this, we used a slightly unusual commercial sample preparation procedure (Swift Accel-NGS 1 S Plus), which allows for the analysis of nicked ssDNA fragments. In contrast to traditional amplicon sequencing applied in DNA data storage, this preparation procedure relies on a proprietary adaptive step and does not require PCR or ligation as a first sample preparation step[19,26]. Thereby, this procedure enables the reading of sequences of incomplete lengths (=fragments). For analysis, the single-strand DNA sequences were aligned and compared based on fragment size. This ensures that synthesis or sequencing errors do not affect the fragment analysis, and the nicking process can be directly evaluated by looking at the fragment-size distribution. Figure 2 shows the fragment-size distribution for a file encoding a 115 kB Jpg image, consisting of 7,373 unique DNA strands for the

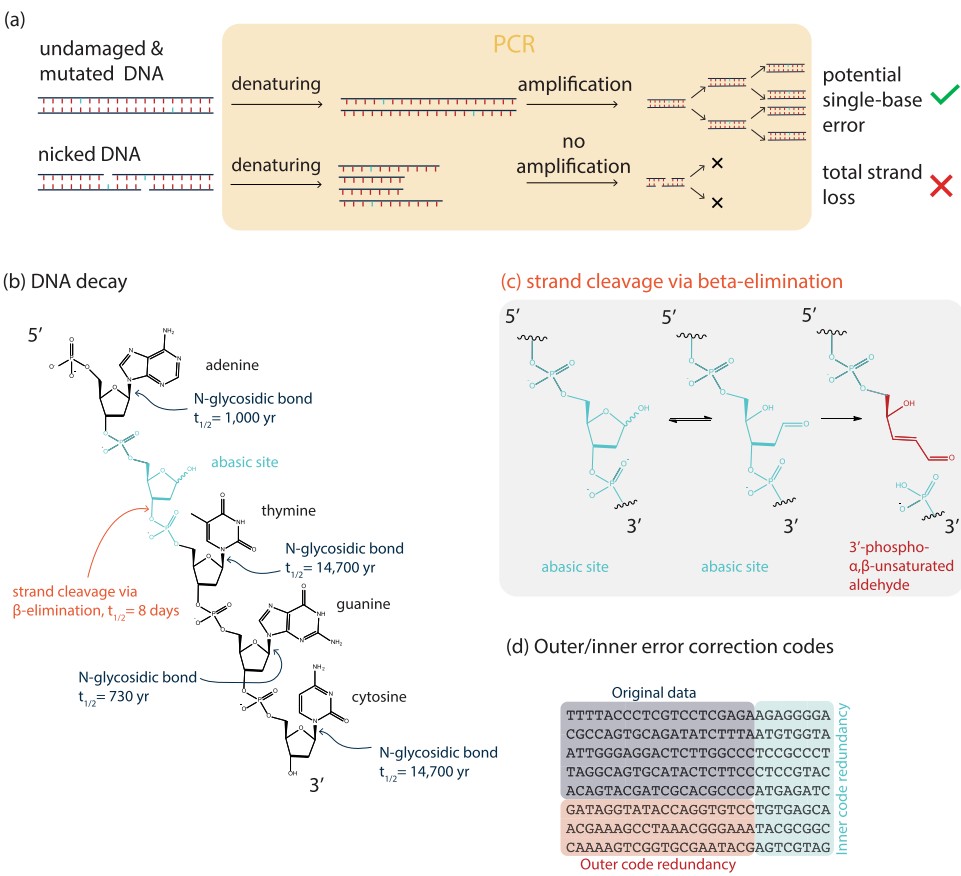

**Fig. 1 Decay pathways during DNA data storage. a** Denaturation of undamaged/mutated DNA and nicked DNA results in potential single-base loss and total strand loss, respectively. **b** Hydrolysis can lead to the release of bases with rates as indicated. Once an abasic site exists, strand cleavage proceeds via β-elimination[3]. **c** Mechanism of β-elimination resulting in a 3'-phospho-α,β-unsaturated aldehyde[33]. **d** Structure of a DNA storage project with outer and inner code[15, 18, 19].

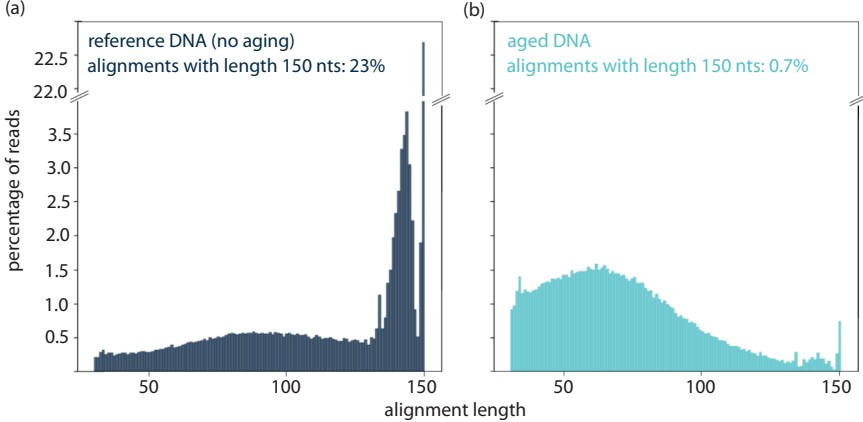

**Fig. 2 Fragment length distribution analysis.** Fragmentation of DNA file (originally containing 7373 unique DNA sequences of length 150 nt). Fragmentation analysis has been conducted by dsDNA denaturation, enzymatic single-strand library preparation and sequencing using Illumina's iSeq100. Subsequently, alignment was performed to obtain the fragment length distribution, excluding sequences with <30 nt length. **a** Reference DNA that was not aged. **b** Sample aged at 30 °C for 6 weeks.

undamaged file (Fig. 2a) and the aged file (Fig. 2b). Per design, the full length of each sequence is 150 nt. We observe that nearly 23% of alignments are of the correct strand length in the file of undamaged DNA, whereas following aging of the sample, only 0.7% of alignments are of the correct strand length. The fragment-size distribution varies as expected: The aged file contains many small fragments between fragment lengths of 50–70 nt and no prevalence of a specific fragment length. This finding

supports the general hypothesis that under well-controlled conditions, DNA decay is a random process[27–30].

Fragmentation of DNA aged at 30 °C shows that nicking results in a sequence length reduction by one-half, approximately. However, no matter how long the resulting fragments are, it may be safely assumed that in a standard sample preparation method involving PCR as initial amplification step[12,13,15,16], these fragments would not amplify (Fig. 1a).

**Table 1 Enzymes used and proposed in various DNA repair formulations.**

| Enzyme | APL repair mix | PreCR ® repair mix | Lindahl et al.[2] |
|---|---|---|---|
| UDG glycosylase | | x | x |
| FPG glycosylase | | x | |
| T4 PDG glycosylase | | x | |
| Endonuclease IV | | x | x |
| Endonuclease VIII | | x | |
| APE1 | x | | |
| RecJ protein | | | x |
| Polymerase I | | | x |
| Bst polymerase | x | x | |
| T4 ligase | | | x |
| Taq ligase | x | x | |

DNA hydrolysis has been firstly investigated by Lindahl[3] showing that guanine and adenine are released from DNA ~20 times faster than cytosine and thymine, without a significant effect from whether the DNA is single-stranded or double-stranded. Once the base is lost (leading to an abasic site in the DNA stand), cleavage via β-elimination is expected to proceed with a half-life of about eight days, leaving the phosphate backbone nicked[24]. Figure 1b summarizes literature DNA decay rates including the rates of release of bases to form apurinic and apyrimidinic sites, and rates of strand cleavage via β-elimination[3,24]. Once an abasic site exists, spontaneous β-elimination proceeds under physiological conditions, resulting in a DNA nick. Even in the absence of water, DNA nicking as a result of an acid-catalyzed and water-independent base loss and subsequent strand breakage has been reported[31].

**Enzyme selection.** As already discussed previously, nature has pathways to repair various types of damage enzymatically. Although damage caused by hydrolysis can be of different chemical morphologies (i.e., resulting in abasic sites, nicks, deaminated cytosine, or fragmentation)[3], our primary interest was to develop a targeted enzymatic repair pathway with the potential to repair the nicks observed in the previous section of this work and thus to ensure that as many DNA strands as possible can be amplified during PCR .

From a most primitive view, one could imagine that a nick comprising a single-base loss (see e.g., Fig. 1a) could be resolved by the simple use of a polymerase and a ligase. The polymerase would replace the missing base, and the ligase would ligate the two portions of the strand. This works very well in many contexts of molecular biology (e.g., Gibson assembly[32]). However, our experiments using such enzyme mixes were not useful to reassembly the nicks caused by hydrolysis. The reason for this can be found if the chemical mechanism of DNA hydrolysis is more closely investigated: After strand cleavage a 3'-phospho-α,β-unsaturated aldehyde is present in the phosphate backbone[33] (Fig. 1c), which is known to obstruct the base-excision-repair pathway[34,35].

There is, however, an enzyme that allows removing the obstructive 3'-phospho-α,β-unsaturated aldehyde. This enzyme is called apurinic/apyrimidinic endonuclease I (APE1) and is most commonly known for its function to incise the phosphate backbone with its endonuclease activity[35,36]. However, APE1 has a functionality to cleave off a 3'phospho-α,β-unsaturated aldehyde (the obstructive group present after DNA nicking) as well, leaving an 3'-OH terminal. This terminal should then be subsequently repaired by the standard base-excision-repair pathway using polymerase and ligase[2,37].

To test the hypothesis that APE1 can remove 3'-phospho-α,β-unsaturated aldehyde groups, we investigated the requirement of the three enzymes (APE1, polymerase, ligase) by individually removing each of the enzymes and quantifying repair using PCR (Fig. 3a), showing that repair mixes without APE1 do not seal the nick in DNA strands. Interestingly, the repair does not depend on the presence of the ligase, which is counterintuitive as polymerases generally cannot seal nicks. However, the employed full-length BST polymerase is a preferred enzyme for nick translation, due to its 5'-3' exonuclease activity[38]. This degrades the displaced strand, and thereby the polymerase can complete the full sequence from the 3' strand of the nick, and the complete dsDNA can be restored without a ligation step[39]. However, this is only possible in conjunction with APE1, which is required for removing the obstructive end induced during degradation. We tested various different polymerases and ligases, as well as the addition of other enzymes, combinations of different enzymes of the same type (i.e., a combination of different endonucleases, polymerases with and without exonuclease activity, and ligases, see Supplementary Figs. 1–3, Supplementary Tables 2 and 3, and Supplementary Method 1) and observed optimal repair activity with Bst polymerase and Taq ligase in combination with APE1 treatment, as quantified by PCR. From our experimental findings, we devised a molecular repair path to seal nicks in DNA strands (Fig. 3b), which highlights the importance of the APE1 enzyme in removing obstructing moieties prior to the polymerase step.

We have further investigated the required repair conditions by tuning the relative amounts of these three enzymes as well as the composition of dNTPs (equimolar mixture of dATP, dTTP, dGTP, and dCTP), NAD+ and ThermoPol® buffer for optimal repair conditions. We optimized the repair time to be 15 min at a repair temperature of 37 °C. We named this mix the "APL repair mix" based on the initials of **A**PE1, **P**olymerase, **L**igase.

**Repair performance.** To visualize the repair process, we denatured a DNA file (file 1 DNA, see Supplementary Table 1 for details) that has been aged at 30 °C for 4 weeks and analyzed the single-strand DNA components using gel electrophoresis. (Fig. 4a). L1 shows reference file 1 DNA (no aging, no repair), L2 shows aged file 1 without repair and L3 shows aged file 1 DNA that has been repaired using APL repair mix. We see that L2 shows barely any band at 150 nt, indicating that only a few strands of the original sequence length (150 nt) are still intact. L3, on the other hand, shows a clear band at 150 nt, indicating that some of the damaged files must have been repaired by APL repair enzymes to the original length of 150 nt. This confirms our hypothesis that the APL repair mix can repair nicks in the phosphate backbone after hydrolytic damage has occurred.

To confirm these qualitative observations, we analyzed the fragment distribution of the APL repaired DNA (similarly to the analysis shown in Fig. 2), including dsDNA denaturation, single-strand library preparation, sequencing using Illumina's iSeq100 and subsequent alignment (Fig. 4b). Whereas we had observed above that the correct sequence length is found in 23% of alignments in the undamaged file and in 0.7% of alignments in the damaged file, results of enzymatic repair show that fragments got repaired, resulting in the file containing 1.7% of alignments with correct sequence length of 150 nt, more than doubling the number of full-length sequences available for decoding. It may be added, that nicked DNA sequences cannot be decoded simply because they lack both PCR primers (Fig. 1a) and would not have been amplified in a standard PCR-based dsDNA sequencing preparation workflow or a random access workflow[15,17]. However, broken sequences pose a second problem as because during fragmentation, sequence indices, which are required for decoding

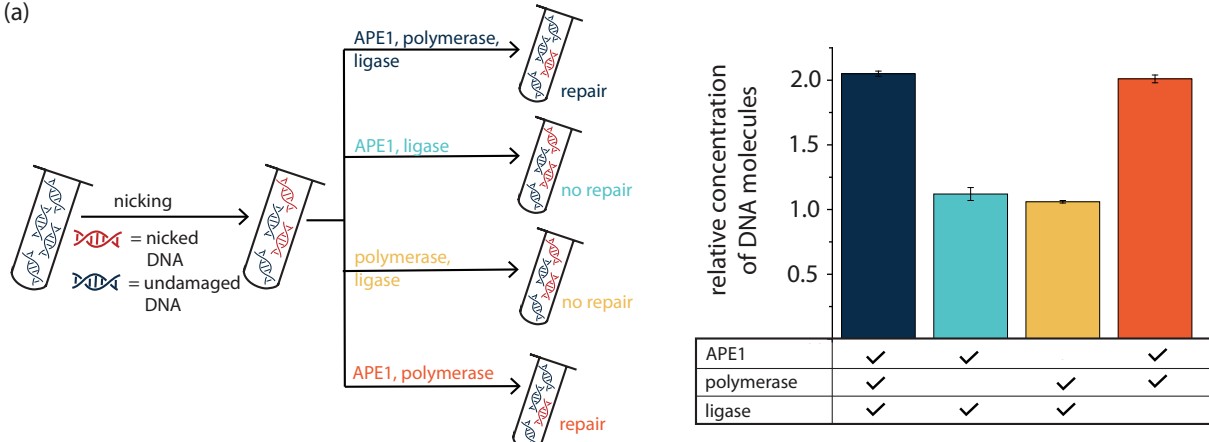

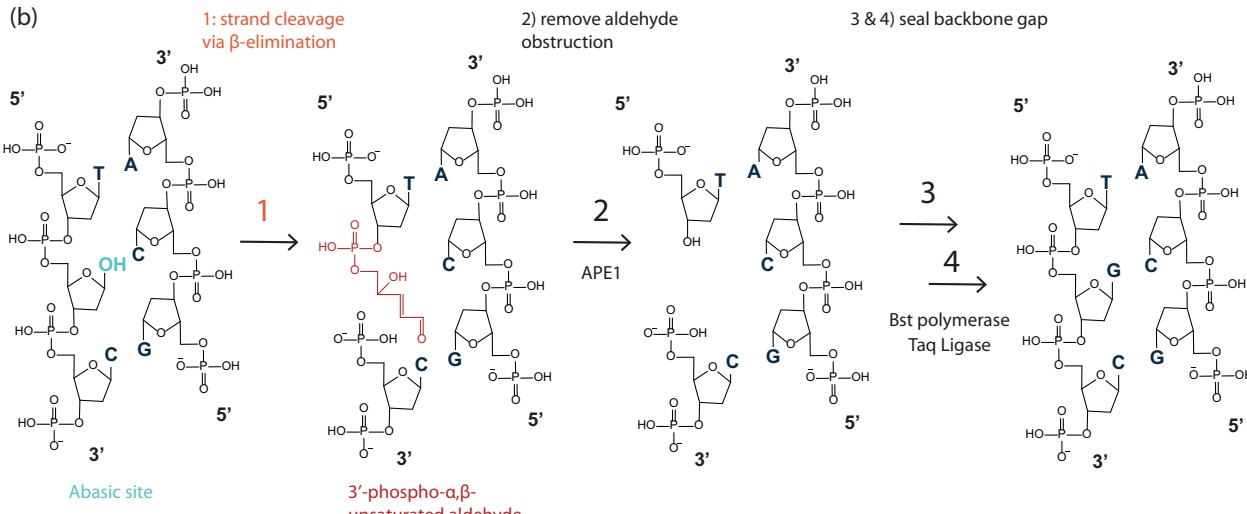

**Fig. 3 Repair pathway of nicked phosphate backbone. a** Quantification of repair potential on aged model DNA (4 weeks, 30 °C), with three enzymes (APE1, Bst Polymerase, Taq ligase), compared to repair results when leaving out either one of the three enzymes during the repair procedure; s.d. from three qPCR replicates, DNA concentration normalized by the nonrepaired sample (rel. conc. of nonrepaired sample = 1.0). **b** Suggested repair pathway of a nick: when the phosphate backbone has been cleaved by β-elimination, APE1 can ingest the 3'-phospho-α,β-unsaturated aldehyde and render the DNA strand with a 3'-OH terminus such that a polymerase can elongate the backbone and insert the missing base. The taq ligase will then seal the strand.

get cutoff[18]. We observe that less shorter fragments (50–70 nt) exist after repair, implying that these sequence lengths were very susceptible for repair.

Using qPCR, we quantified the repair performance of the APL repair mix by comparison with a commercial enzyme mix, marketed for general-purpose DNA repair (PreCR®). As in a DNA data-storage setting, PCR is the first process performed following storage, the amount of amplifiable DNA reported in Fig. 4c is the main performance metric for DNA repair in this application. The figure shows the percentage of intact DNA per DNA pool investigated (model DNA and file 1 DNA described in Supplementary Table 1) without damage (assumed to be 100% intact), after damage, and after repair. We observe that the self-devised APL mix shows repair of up to 31% of DNA sequences that would have otherwise been lost (=ratio of repaired DNA over lost DNA). In addition, we observed that the less severe the damage was, the more relative repair was achieved. For example, after damaging model DNA, 42% of DNA strands were still intact. Repair, however, recovered DNA strands from the pool, increasing the percentage of intact DNA strands to 73% (i.e., 53% of the damaged sequences could be repaired). On the other hand, after

aging file 1 DNA, only 5% of DNA strands were still amplifiable. With repair, this fraction was improved by more than threefold to 16%. DNA repair was also effective using the commercial enzyme mix. The performance of the PreCR mix (an increase of 5–13% amplifiable DNA for file 1, see Fig. 4c) was slightly lower than the performance of the APL enzyme mix, developed specifically for repair of hydrolytic damage of synthetic oligo pools.

In combination, gel electrophoresis, fragmentation, and qPCR analyses show that repair of fragmented synthetic DNA is successful under optimized repair conditions. We observe that the less damage a file has seen, the more repair is possible. This is expected, as during prolonged damage, a single sequence may be nicked several times, making repair more and more challenging by the decreased melting points of the individual fragments (Supplementary Fig. 4 and Supplementary Note 1). Fragment analysis has additionally shown that repair results in an increase in sequences with the correct sequence length, which is an important indicator for whether or not a file can be amplified and sequenced.

As previously mentioned, commercial enzymatic repair mixes exist (i.e., PreCR® repair mix) and can be bought as complete

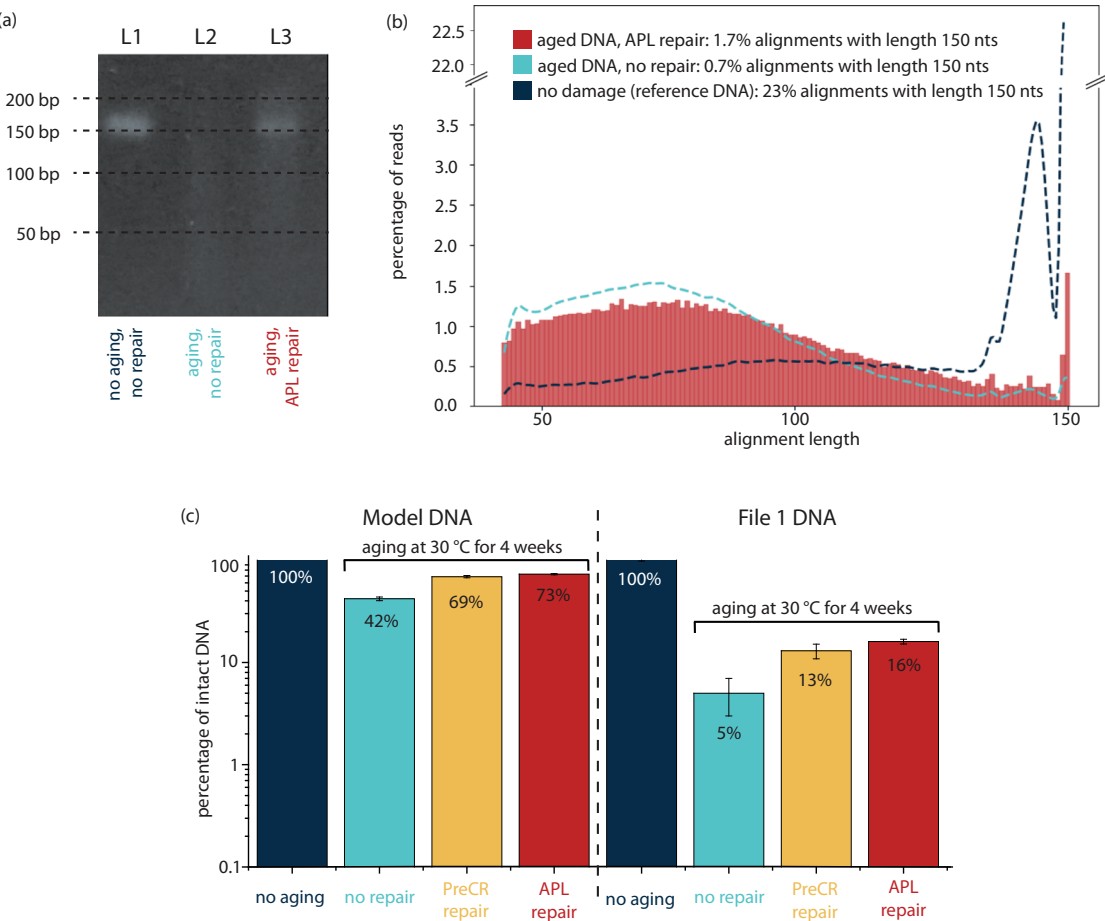

**Fig. 4 Enzymatic repair of DNA libraries. a** Denaturing gel of file 1 DNA aged at 30 °C for 4 weeks. Bands show reference DNA (no aging, no repair), damaged DNA without repair (aging, no repair) and damaged DNA that has been treated with APL repair (aging, APL repair). **b** Fragmentation of file 1 DNA after alignment comparing reference and aged DNA to APL repaired DNA. **c** Repair potential of commercial enzymatic repair mix (PreCR repair) and self-composed enzymatic repair mix (APL repair) shown for damaged files aged at 30 °C for 4 weeks. Data is shown on a log scale due to the exponential nature of qPCR; s.d. from three qPCR replicates.

---

**Box 1 ▌ Comparison of repair mixes**

As referred to throughout this work, different enzymatic repair pathways exist (commercially or theoretically) that allow for the repair of compromised DNA strands. Table 1 lists the components of the commercially available PreCR® repair mix, the enzymes as suggested by Lindahl et al. for the base-excision-repair pathway[2], and our self-composed APL repair mix. Repairing the nick that occurred during hydrolysis does not require the full procedure of the base-excision-repair pathway, as the repair of mutations is not necessary for data recovery for DNA data storage (the error-correction code can easily account for base mutations). This is why the APL repair mix requires only three enzymes that chemically target the obstructive damage caused by nicking.

---

enzymatic kits. Such enzymatic kits have already been investigated for forensic applications, the recovery of ancient DNA or cell-free DNA in maternal plasma for prenatal testing, where compromised DNA samples were repaired using the PreCR® repair kit[11,40,41]. Repair results, however, varied across applications: Treating ancient DNA with the PreCR® enzyme kit generally showed no appreciable repair[10]. Limited functionality of repair kits was also observed across most studies of forensic DNA. Some of the most promising results were shown when trying to increase the number of detectable alleles in impaired DNA samples[11,40–45].

To our best knowledge, for the purpose of repairing synthetic DNA strands, there exists no previous work that has investigated the application of commercial enzymatic repair mixes. In addition to showing that DNA repair is useful in reverting damage in a DNA storage application using the PreCR® repair

enzymes mix, we have developed a problem-focused, more cost-effective, and higher-performing enzyme mix (APL). Due to the lower complexity of the APL repair mix (see Box 1), it was drastically more cost-effective than the PreCR® mix (Supplementary Table 4). We subsequently quantified the repair of the two approaches using PCR (Fig. 4c) and observed that repair capacity of the PreCR® repair mix in synthetic DNA is 10–25% lower than APL repair across the investigated file sizes and degrees of damage. In the investigated application scenario of DNA data storage, the added cost of DNA repair after storage (ca. 0.3 USD per reaction) appears small in contrast to the cost of synthesizing more redundant DNA prior to storage. This is especially relevant as the cost of DNA synthesis remains high, at about 1000 USD/MB[19], and consequently every 1% of added redundancy would add 10 USD of cost per MB stored. The trade-offs between DNA redundancy and repair cost is discussed in the following section.

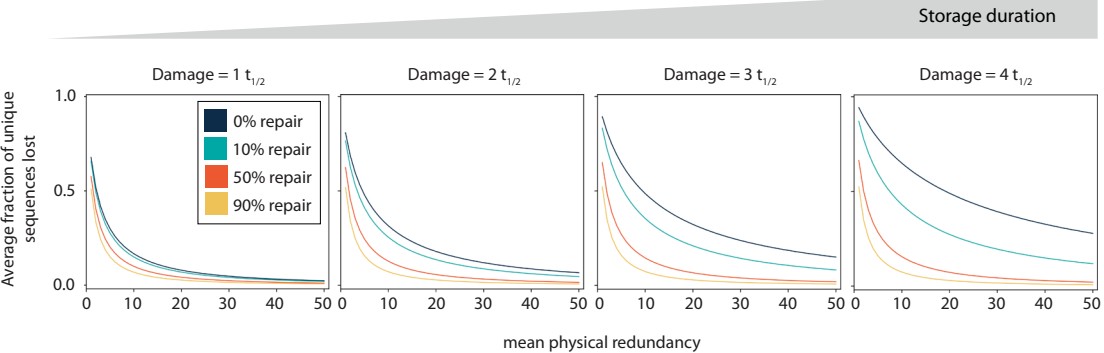

**Fig. 5 Simulating damage and repair for high-density storage regimes.** For a given physical redundancy and storage horizon (expressed in half-lives of DNA), the data give the fraction of unique sequences lost from the pool during storage (dark blue lines). This loss of sequences has to be compensated by the logical redundancy of an error-correcting code. With DNA repair enabled, a fraction of the lost sequences can be recovered, and less logical redundancy is required. The benefits of DNA repair is the largest in applications with a low physical redundancy and long storage durations, which are target domains for DNA data storage.

**Simulating damage and repair for information recovery.** Whereas our experimental work has focused on files with high copy numbers of DNA strands (>350-fold physical coverage, see Supplementary Table 1), the future of DNA data storage aims at decreasing the volume required for storage, and thus increasing the data density. The physical limit of reliable DNA data retrieval was calculated to be up to 17 exabytes per gram, at only around tenfold physical coverage[46].

Potential future applications for DNA data storage would, for example, be the archiving of valuable information in national or state archives. Whereas on the one hand the density for storage is desired to be high (to grasp the full potential of DNA as a medium for data storage), it is imperative that data can still be read subsequently to storage. As the effect of physical redundancy, error-correction coding redundancy (logical redundancy), and damage/repair of sequences is complex, we have simulated damage and enzymatic repair of DNA libraries for low physical coverage (= high storage density) scenarios (1–50), to express the benefits information recovery can have.

The design of a DNA storage starts with defining the targeted information density and storage horizon, resulting in a targeted physical redundancy and expected DNA decay. Figure 5 shows the expected loss of unique sequences (i.e., unique sequences no longer present in the pool), from a simulation (see full code in Supplementary Note 2) as a function of these parameters, thereby accounting for the expected effects of synthesis bias and PCR bias[20,47]. An error-correction code would then be tuned to compensate for these anticipated losses by adding appropriate logical redundancies to the original data. As an example close to the current state of the art, data would be stored at a mean physical redundancy of 10 for a storage horizon of 1000 years at room temperature (equivalent to $t_{1/2} = 2$ for DNA stored in silica encapsulates, for which hydrolytic damage is also the most prominent decay factor[16]). Under these conditions, the simulation (Fig. 5) predicts that an error-correcting code needs to compensate for a loss of ca. 30% of the unique sequences in the absence of enzymatic repair. According to the Shannon limit this requires a logical redundancy of at least 30%, and consequently, a surplus of 43% of unique sequences would have to be synthesized (surplus = $(1/(1-0.3) - 1)*100$), to allow for an error-less data recovery. However, when applying a DNA repair strategy, originally lost sequences can be recovered, as shown experimentally in Fig. 4c. Assuming a repair capable of recovering 50% of the sequences lost (=ca. average experimental performance of the APL mix) at these parameter choices (phys red. = 10, $t_{1/2} = 2$), much less sequences would be missing from the pool (12%),

requiring a drastically smaller logical redundancy overhead, and a lower number ($13.5\% = 1/(1-0.12) - 1*100$) of surplus DNA sequences. In this scenario this leads to a considerable cost reduction of ca. 200 USD/MB as DNA synthesis costs (ca. 1000 USD/MB)[19] is currently the main bottleneck of DNA data storage. Consequently, the application of the rather straightforward APL enzymatic repair (<0.3 USD/reaction, Supplementary Table 4) applied at the end of the storage horizon and directly prior to data readout, has great impact on the synthesis cost and readout reliability of the data. Being able to read information from low physical redundancy pools is also of importance, if the DNA pool is distributed, as in the case for DNA of things[48] and enzymatic repair could also be used to recover data from DNA data-storage systems, in which DNA has been stored longer than anticipated.

## Discussion

In this work, we analyze the decay and repair process of synthetic DNA for DNA data-storage applications. We present a molecular analysis of the chemical process during nicking as a form of DNA decay and focus on nicking due to hydrolysis. Experimental conditions represent real-life scenarios during DNA handling steps and illustrate the threat of water (in the form of a medium for handling, and in the form of humidity during storage). We find that hydrolysis causes molecular obstruction at the 3' terminal when the strand is nicked, which results in strand loss during the denaturation steps of PCR. As strand loss can only be compensated by redundancy of additional strands, nicking is a very expensive form of damage, compared to single-nucleotide mutations. We have thus devised an enzymatic solution containing three single enzymes (APE1, Bst polymerase, and Taq ligase) targeted at removing the obstructive aldehyde group after nicking and subsequently sealing the nick to repair DNA strands such that PCR amplification is possible again. We find that repair can recover information, with higher recovery in low-damage regimes and show more than 30% recovery. Comparing our self-devised repair mix to commercial kits (i.e., PreCR® repair mix), our repair results in cheaper, faster, and more effective recovery of DNA.

At low physical coverage regimes, losing DNA strands can lead to data (contained in that strand) being lost and compromised readout. We have thus simulated the requirement of physical and logical redundancy to predict and control the encoding process for DNA data storage such that readout is still possible. This will be especially useful when moving towards high-density storage, for which DNA is especially well suited.

Enzymatic repair of DNA can, in the future, be especially interesting to data archives, when converting traditional archiving processes to include DNA as a storage medium becomes of greater interest. When storage times are unknown, a method for information recovery is beneficial, offering a recovery option if standard decoding and readout shall fail.

## Methods

**PreCR enzyme kit—optimal repair conditions**. The optimal ratios for repair are as follows. Master mix (ThermoPol buffer, dNTP, NAD + (5:2:1)) was added to DNA (1 ng/μL) and water/PreCR® repair enzymes (ratio: 4:25:1, respectively). Enzymes were added last, and reactions were kept on ice. After the addition of all components, reaction vials were shaken by hand and centrifuged. Repair was carried out in 30 °C incubator for 4 h and subsequently centrifuged again. PCR quantification was performed by diluting 1 μL of incubated DNA in 499 μL water.

**Self-composed enzyme repair mix—optimal repair conditions**. Individual components were added on ice to make master mix: (ThermoPol buffer, dNTP, NAD+, APE1, Bst polymerase (full length), Taq ligase (ratio: 10:4:2:1:1:1). For repair reactions, 18.8 μL mili-Q water, 6.25 μL DNA (1 ng/μL), and 1.57 μL master mix were added (for control samples, the 1.57 μL master mix were exchanged for water). After the addition of all components, reaction vials were shaken by hand and centrifuged. Repair was carried out in 30 °C incubator for 15 min. Samples were subsequently centrifuged again. PCR quantification was performed by diluting 1 μL of incubated DNA in 499 μL water.

**Sequencing preparation**. Damaged, repaired and reference samples were denatured and prepared for sequencing using the Swift Biosciences Accel-NGS® 1S Plus DNA Library Kit and following the manufacturer's recommended protocol. To preserve small DNA fragments to the best extent possible, small fragment retention was performed according to the manufacturer's instructions. All steps were performed separately for all samples, and no nicked samples were mixed.

**Denaturing gel**. DNA samples were denatured in the following way and then run on normal Agarose 2% SYBR Gold II gels: 200 mM sodium phosphate solution (pH 7.0) was prepared by adding 1.64 g of $Na_2HPO_4$ and 1.02 g of $NaH_2PO_4$ to 80 mL of deionized water. The pH was adjusted to 7 by using NaOH (this step may not be necessary). Water was added to bring the total volume to 100 mL. Denaturing buffer was made by adding 200 mM sodium phosphate solution (23.8 μL), 40% glyoxal (83.3 μL), and 99.7% DMSO (238 4 μL).

**Statistics and reproducibility**. For qPCR, analysis was performed in triplicate. qPCR analysis was performed for two sample sets (model DNA & File 1 DNA). No data were excluded.

**Reporting summary**. Further information on research design is available in the Nature Research Reporting Summary linked to this article.

## Data availability

Source data underlying figures are presented in Supplementary Data 1. The sequencing data underlying Fig. 4 is publicly available at https://figshare.com/articles/dataset/Sequencing_data/21070684.

## Code availability

The code utilized to simulate the losses of sequences as a function of physical redundancy and storage time can be found in the Supplementary Information.

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

## Acknowledgements

The authors thank Microsoft for funding and the Beat Christen Group for giving access to the iSeq100 sequencer.

## Author contributions

R.N.G. and K.S. initiated the project, R.N.G., K.S., B.H.N., and W.J.S. supervised the project. R.H., L.C.M., and B.H.N. performed data analysis. L.C.M, G.L., and W.D.C. performed laboratory experiments, T.D. and A.L.G. coded, implemented, and analyzed the model data. All figures prepared by L.C.M. L.C.M. and R.N.G. wrote the manuscript with input and approval from all authors.

## Competing interests

R.N.G. and W.D.C. are authors of a patent application licensed to Microsoft. K.S. and B.H.N. are Microsoft employees. The remaining authors declare no competing interests.
