## [Peer Review File · Communications Biology]

Reviewers' comments:

Reviewer #1 (Remarks to the Author):

In this work, Meiser et al. describe an enzyme cocktail that can be used to improve data recovery from DNA stored data. Specifically, they show that APE1 is required to repair the resulting b-aldehyde from beta elimination of an abasic site; quantitatively characterized the repair efficiency of their cocktail using qPCR and iSeq100 analysis; and benchmarked their system to a commercial DNA repair kit, PreCR. They show that the APL cocktail repair efficiency is dependent upon the original extent of DNA damage and that they can improve the repair efficiency of up to ~30%. Lastly, they simulated the amount of redundancy and repair needed to achieve high-density/high efficiency DNA storage. Findings from this work has applications in DNA storage, but more broadly to any DNA retrieval application (i.e., ancient DNA, prenatal DNA testing, exc.). Upon addressing minor revisions, recommend publication.

Specific Comments:

Line 61: The authors emphasize cost and DNA synthesis being the major bottle neck. What exactly is the cost increase (i.e., 20 ng scale synthesis versus 150 ng scale synthesis from IDT) and please expand upon why synthesis is the main bottle neck. For example, what are the steps for DNA storage and retrieval, and why is synthesis the bottle neck, time?

Figure 1 and Line 81: Suggest reformatting figure to switch panel C to panel A, since this is discussed first in the text. Also should reference citation #3 in figure description.

Line 106: Write out file sizes. Also, reference exactly where in supplemental (i.e., Supplemental Figure # or Supplemental Table #...)

Supplemental Table 1: What is average copy # for each set of DNA? Please include in table.

Line 113 and 115: Write out what is unique. Is it ligation versus PCR?

Line 117: 'analysis, the single strand DNA sequences were aligned' and compared based on fragment size.

Line 177: Did you test polymerases with 3' exonuclease activity?

Figure 3. Please include negative control of no enzymes. Also, what DNA is this based off of (modle or DNA 1). Because authors show that % degradation is critical on repair efficiency and the two different DNA have different degradation.

Line 184: Reference exact supplemental figure.

Line 210: Shouldn't this be 31%?

Paragraph of line 240: Should discuss first before Line 214. Should talk about this first (qualitative observation), then go into the quantitative information (above paragraph).

Line 251: Reference where discussed in supplemental

Line 268: Describe cost per reaction. How does this compare to synthesizing more DNA?

Line 271: How much lower? Please include in text.

Line 274: What is the copy #? Please include in text.

Line 284: What is the coverage? Please include in text.

Line 302: How does this compare to APL repair?

Line 335: Can you explain in more detail how your simulation was made, specifically how the % repaired is incorporated into your simulation?

DNA synthesis cost -> DNA synthesis costs

Acknowledgements: than -> 'thank'

Reviewer #2 (Remarks to the Author):

In this manuscript, the authors proposed an enzymatic repair system for synthetic DNA used in data storage. They first provided a detailed problem statement on the negative impacts of DNA nicking and quantified such decay through sequencing. Based on the damage mode, they designed an enzyme mix specifically for their hypothesized repair pathway and measured its performance with different methods. Their results showed that the APL repair mix outperformed a commercially available DNA repair kit and effectively repaired nicked DNA in DNA libraries. Finally, they simulated the unique sequence lost in DNA storage over time and showed that applying DNA repair could enable low physical coverage with reduced cost and improved read-out reliability.

I enjoy reading this paper as it is well-written with a clear, smooth logic flow. The experiments are carefully designed, executed, and presented. The work described here is novel and would be stimulating for a broad spectrum of readers interested in DNA data storage and other relevant technologies. However, there are a few points that might help improve the clarity of this manuscript and should be addressed before its publication:

1. The first figure panel cited in the main text is Fig. 1c in Line 81. It might help the readers follow better if it was rearranged and labeled as Fig. 1a.
2. Line 87: Could the authors provide more explanation on the cost and symbol? A simple diagram might help.
3. Fig. 3a: The authors should discuss more the data from APE1 + polymerase. Why does it look comparable to the data from the APL mix?
4. Line 201: The authors should briefly introduce the PreCR kit in this paragraph.
5. Fig. 4a: Is there a specific reason for having the y axis in the log scale? It is not so easy for the readers to visualize the differences between bars and the errors.
6. Line 270 and Fig. 4b-c: The authors should also include gel and fragmentation data from PreCR repair here (or in the supplementary information) and provide a fair benchmarking.
7. Line 286: It is unclear how the simulation parameters here echo the experimental work in previous sections. Is there water in the silica encapsulation mentioned? Is the DNA repair only applied at the end of the entire period of storage/damage? A few extra details relevant to the experimental results might help the readers grasp the concept better.

Reviewer #1 (Remarks to the Author):

In this work, Meiser et al. describe an enzyme cocktail that can be used to improve data recovery from DNA stored data. Specifically, they show that APE1 is required to repair the resulting b-aldehyde from beta elimination of an abasic site; quantitatively characterized the repair efficiency of their cocktail using qPCR and iSeq100 analysis; and benchmarked their system to a commercial DNA repair kit, PreCR. They show that the APL cocktail repair efficiency is dependent upon the original extent of DNA damage and that they can improve the repair efficiency of up to ~30%. Lastly, they simulated the amount of redundancy and repair needed to achieve high-density/high efficiency DNA storage. Findings from this work has applications in DNA storage, but more broadly to any DNA retrieval application (i.e., ancient DNA, prenatal DNA testing, exc.). Upon addressing minor revisions, recommend publication.

We would like to thank the referee for the very concise and precise summary of our article, and the positive conclusion. We agree that the work has implications to other applications of DNA, and hope that this work on purely synthetic DNA can be used as a basis for scientists working with natural DNA samples, and therefore less controlled storage conditions.

Specific Comments:

Line 61: The authors emphasize cost and DNA synthesis being the major bottle neck. What exactly is the cost increase (i.e., 20 ng scale synthesis versus 150 ng scale synthesis from IDT) and please expand upon why synthesis is the main bottle neck. For example, what are the steps for DNA storage and retrieval, and why is synthesis the bottle neck, time?

It should be understood that the challenge faced not moving from a 20 ng scale to a 150 ng scale (i.e. more physical redundancy), but that the synthesis of more individual oligos is necessary if errors have to be accounted for. To make this more clear to the reader, and in line with comments of the referee further below, the discussion of the synthesis cost has been deepened:

This does not mean that the DNA pools have to be synthesized at a larger scale (i.e. having more copies per oligo), but that more unique oligos have to be synthesized to enable error correction (e.g. larger outer code redundancy).^{15,18,19} The cost of this scales directly with the number of expected errors.²⁰

Figure 1 and Line 81: Suggest reformatting figure to switch panel C to panel A, since this is discussed first in the text. Also should reference citation #3 in figure description.

We thank the referee for the useful improvement, which we have implemented as advised.

Line 106: Write out file sizes. Also, reference exactly where in supplemental (i.e., Supplemental Figure # or Supplemental Table #...)

Done

Supplemental Table 1: What is average copy # for each set of DNA? Please include in table.

As requested, this information was added to the table. We would like to add that with all analysis performed in this work, the physical coverage is not expected to affect the results in any way.

Supplementary Table 1: DNA used for this work. Shown are properties of each set of DNA

DNA name	Encoded information	Size of file [bytes]	Size of file [unique DNA strands]	Length of DNA strands	Physical coverage [copies of file used per analysis]
Model DNA	-	-	1	113	4,000,000
File 1 DNA	Jpg image	115,394	7,373	150	386

Line 113 and 115: Write out what is unique. Is it ligation versus PCR?

The difference here is not ligation vs PCR (the two standard technologies used for PCR sample prep). In contrast, this is a proprietary commercial technology for the sequencing of single strands, neither requiring PCR nor ligation as first sample prep step. The text in the manuscript was adapted to better explain this:

For this we used a slightly unusual commercial sample preparation procedure (Swift Accel-NGS 1S Plus), which allows for the analysis of nicked ssDNA fragments. In contrast to traditional amplicon sequencing applied in DNA data storage, this preparation procedure relies on a proprietary adaptase step and does not require PCR or ligation as a first sample preparation step.^{19,27}

Line 117: ‘analysis, the single strand DNA sequences were aligned’ and compared based on fragment size.

corrected ,thank you!

Line 177: Did you test polymerases with 3’ exonuclease activity?

yes, see Supplementary Figure 1. Phi29 has exonuclease activity, and did not perform very well. To better guide the reader the relevant paragraph was improved to:

We tested various different polymerases and ligases, as well as the addition of other enzymes, combinations of different enzymes of the same type (i.e., a combination of different endonucleases, polymerases **with and without exonuclease activity, and** ligases) and observed optimal repair activity with Bst polymerase and Taq ligase in combination with APE1 treatment, as quantified by PCR (see Supplementary **Information Figure 1**).

Figure 3. Please include negative control of no enzymes. Also, what DNA is this based off of (modle or DNA 1). Because authors show that % degradation is critical on repair efficiency and the two different DNA have different degradation.

The missing information was added to the figure caption. All data is compared to the negative control without the use of enzyme. This is now also noted more clearly in the figure caption.

Figure 3: Repair pathway of nicked phosphate backbone. (a) Quantification of repair potential **on aged model DNA (4 weeks, 30°C)**, with three enzymes (APE1, Bst Polymerase, Taq ligase), compared to repair results when leaving out either one of the three enzymes during the repair procedure; s.d. from 3 qPCR replicates, **DNA concentration normalized by the non-repaired sample (rel. conc. of non repaired sample = 1.0)**. (b) suggested repair pathway of a nick: When the phosphate backbone has been cleaved by β -elimination, APE 1 can ingest the 3'-phospho- α,β -unsaturated aldehyde and render the DNA strand with a 3'-OH terminus such that a polymerase can elongate the backbone and insert the missing base. The taq ligase will then seal the strand.

Line 184: Reference exact supplemental figure.

This was implemented, thank you!

Line 210: Shouldn't this be 31%?

The calculation in the manuscript is correct, as the number given is the ratio of repaired DNA (31%) over the amount of lost DNA ($100\% - 42\% = 58\%$), yielding ($31/58 = 53\%$). To make this clear to the reader, the following was added to the manuscript:

We observe that the self-devised APL mix shows repair of up to 31% of DNA sequences that would have otherwise been lost **(=ratio of repaired DNA over lost DNA)**.

Paragraph of line 240: Should discuss first before Line 214. Should talk about this first (qualitative observation), then go into the quantitative information (above paragraph).

This is good input the text was correspondingly rearranged, starting with the most qualitative observation (gel electrophoresis) and completing with the most quantitative analysis (qPCR). Figure 4 was also rearranged to account for this change.

Figure 4: Enzymatic repair of DNA libraries. (a) denaturing gel of file 1 DNA aged at 30 °C for 4 weeks. Bands show reference DNA (no aging, no repair), damaged DNA without repair (aging, no repair) and damaged DNA that has been treated with APL repair (aging, APL repair). (b) Fragmentation of file 1 DNA after alignment comparing reference and aged DNA to APL repaired DNA. (c) repair potential of commercial enzymatic repair mix (PreCR repair) and self-composed enzymatic repair mix (APL repair) shown for damaged files aged at 30 °C for 4 weeks. **Data is shown on a log scale due to the exponential nature of qPCR;** s.d. from 3 qPCR replicates.

Line 251: Reference where discussed in supplemental

Link to supporting information completed.

Line 268: Describe cost per reaction. How does this compare to synthesizing more DNA?

There is no easy answer to this question, as this depends on several factors. We have discussed the requirement of more DNA synthesis in the following section, together with its dependencies (Simulating damage and repair). To make this clearer to the reader, the section has been reworded as follows:

In addition to showing that DNA repair is useful in reverting damage in a DNA storage application using the PreCR® repair enzymes mix, we have developed a problem focused, more cost effective and higher performing enzyme mix (APL). Due to the lower complexity of the APL repair mix (see Box 1), it was significantly more cost effective than the PreCR® mix (Supporting Info Table 4). We subsequently quantified repair of the two approaches using PCR (Fig. 4c) and observed that repair capacity of the PreCR® repair mix in synthetic DNA is 10 - 25 % lower than APL repair across the investigated file sizes and degrees of damage. In the investigated application scenario of DNA data storage, the added cost of DNA repair after storage (ca. 0.3 USD per reaction) appears small in contrast to the cost of synthesizing more redundant DNA prior to storage. This is especially relevant as the cost of DNA synthesis remains high, at about 1000 USD / MB,²⁵ and consequently every 1 % of added redundancy would add 10 USD of cost per MB stored. The trade-offs between DNA redundancy and repair cost is discussed in the following section.

Line 271: How much lower? Please include in text.

Information (10 - 25%) added to text.

Line 274: What is the copy #? Please include in text.

Information (> 350 fold physical redundancy) added to text

Line 284: What is the coverage? Please include in text.

Missing information (1-50 fold) added to text.

Line 302: How does this compare to APL repair?

The text was changed to include cost information:

Consequently, the application of the rather straightforward APL enzymatic repair (< 0.3 USD/reaction, Supporting Info Table 4) applied at the end of the storage horizon and directly prior to data readout, has significant impact on the synthesis cost and read out reliability of the data.

and

In this scenario this leads to a significant cost reduction of ca. 200 USD / MB as DNA synthesis costs (ca. 1000 USD /MB)²⁵ is currently the main bottleneck of DNA data storage.

Line 335: Can you explain in more detail how your simulation was made, specifically how the % repaired is incorporated into your simulation?

The text was modified to better explain this. In addition, the full code is given in the Supporting Information.

However, when applying a DNA repair strategy, originally lost sequences can be recovered, as shown experimentally in Figure 4c. Assuming a repair capable of recovering 50% of the sequences lost (= ca. average experimental performance of the APL mix) at these parameter choices (phys red. = 10, $t_{1/2}=2$), significantly less sequences would be missing from the pool (12%), requiring a drastically smaller logical redundancy overhead, and a lower number ($13.5\% = 1/(1-0.12)-1*100$) of surplus DNA sequences. In this scenario this leads to a significant cost reduction of ca. 200 USD / MB as DNA synthesis costs (ca. 1000 USD /MB)²⁵ is currently the main bottleneck of DNA data storage.

DNA synthesis cost -> DNA synthesis costs

Acknowledgements: than -> 'thank'

Thank you, these errors have been fixed!

Reviewer #2 (Remarks to the Author):

In this manuscript, the authors proposed an enzymatic repair system for synthetic DNA used in data storage. They first provided a detailed problem statement on the negative impacts of DNA nicking and quantified such decay through sequencing. Based on the damage mode, they designed an enzyme mix specifically for their hypothesized repair pathway and measured its performance with different methods. Their results showed that the APL repair mix outperformed a commercially available DNA repair kit and effectively repaired nicked DNA in DNA libraries. Finally, they simulated the unique sequence lost in DNA storage over time and showed that applying DNA repair could enable low physical coverage with reduced cost and improved read-out reliability.

I enjoy reading this paper as it is well-written with a clear, smooth logic flow. The experiments are carefully designed, executed, and presented. The work described here is novel and would be stimulating for a broad spectrum of readers interested in DNA data storage and other relevant technologies. However, there are a few points that might help improve the clarity of this manuscript and should be addressed before its publication:

We thank the author for the very positive assessment of our work. Please find our responses and actions taken to the individual questions raised.

1. The first figure panel cited in the main text is Fig. 1c in Line 81. It might help the readers follow better if it was rearranged and labeled as Fig. 1a.

This was also requested by referee 1, and was consequently implemented (see adapted figure below)

2. Line 87: Could the authors provide more explanation on the cost and symbol? A simple diagram might help.

Also, this point was raised by referee 1, a corresponding figure part was added to figure 1

Figure 1: Decay pathways during DNA data storage. (a) Denaturation of undamaged/mutated DNA and nicked DNA results in potential single-base loss and total strand loss, respectively. (b) Hydrolysis can lead to the release of bases with rates as indicated. Once an abasic site exists, strand cleavage proceeds via β -elimination.³ (c) Mechanism of β -elimination resulting in a 3'-phospho- α,β -unsaturated aldehyde.²⁶ (d) structure of a DNA storage project with outer and inner code.^{15,18,19}

3. Fig. 3a: The authors should discuss more the data from APE1 + polymerase. Why does it look comparable to the data from the APL mix?

Indeed, the performance of APE1 + polymerase is good, and nearly comparable to the APL mix. At first this seems counterintuitive. However, this is well in line with the 5'-3' exonuclease activity of the full length BST polymerase, which is also a preferred nuclease for nick translation. To explain this to the reader, the following was added to the manuscript:

Interestingly, the repair does not depend on the presence of the ligase, which is counter intuitive as polymerases generally cannot seal nicks. However, the employed full-length BST polymerase is a preferred enzyme for nick translation, due to its 5'-3' exonuclease activity.³⁸ This degrades the displaced strand, and

thereby the polymerase can complete the full sequence from the 3' strand of the nick and the complete dsDNA can be restored without a ligation step.³⁹ However, this is only possible in conjunction with APE1, which is required for removing the obstructive end induced during degradation.

4. Line 201: The authors should briefly introduce the PreCR kit in this paragraph.

We thank the author for noting this lack of introduction, and have included the following text:

Using qPCR, we quantified the repair performance of the APL repair mix by comparison with a commercial enzyme mix, marketed for general purpose DNA repair (PreCR®).

5. Fig. 4a: Is there a specific reason for having the y axis in the log scale? It is not so easy for the readers to visualize the differences between bars and the errors.

Yes, there are two reasons for the log scale: data collection is performed by quantitative PCR, which in its core is an exponential method. Consequently, the experimental errors also originate from an exponential signal (CT). In addition, the use of the logarithmic scale enables us to better visualize the results of the File 1 experiment, which would otherwise be hidden close to the x-axis. To ensure that the reader is not misled by the logarithmic axis, the actual values are added to the bars. In order to further ensure that the reader is not misled, the following was added to the figure caption:

Data is shown on a log scale due to the exponential nature of qPCR

6. Line 270 and Fig. 4b-c: The authors should also include gel and fragmentation data from PreCR repair here (or in the supplementary information) and provide a fair benchmarking.

We would like to point out, that qPCR allows a full benchmarking of the enzyme performance and consequently no sequencing of the comparative enzyme mixes was performed. In most common DNA data storage procedures, DNA is amplified following storage (either by common primers or by random access primers). Therefore, the measure of amplifiable DNA by qPCR directly relates to the performance of the repair step. The analysis of the fragment length was performed to better visualize the repair step. To make this more clear to the reader, the following clarifications were added to the manuscript. Also, as proposed by referee 1, the order of the presentation of the analysis was reverted, with the quantitative data from the qPCR analysis now being presented last.

DNA repair was also effective using the commercial enzyme mix. The performance of the PreCR mix (increase of 5% to 13% amplifiable DNA for file 1, see Fig.4c) was slightly lower than the performance of the APL enzyme mix, developed specifically for repair of hydrolytic damage of synthetic oligo pools.

and

To visualize the repair process, we denatured a DNA file (file 1 DNA, see supporting Info for details) that has been aged at 30 °C for 4 weeks and analyzed the single strand DNA components using gel electrophoresis. (Fig. 4a).

and

As in a DNA data storage setting, PCR is the first process performed following storage, the amount of amplifiable DNA reported in Fig. 4c is the main performance metric for DNA repair in this application.

7. Line 286: It is unclear how the simulation parameters here echo the experimental work in previous sections. Is there water in the silica encapsulation mentioned? Is the DNA repair only applied at the end of the entire period of storage/damage? A few extra details relevant to the experimental results might help the readers grasp the concept better.

Following the referees comment, we have included the following text to improve the connection between the experimental and simulation part of the manuscript:

As an example close to the current state of the art, data would be stored at a mean physical redundancy of 10 for a storage horizon of 1000 years at room temperature (equivalent to $t_{1/2} = 2$ for DNA stored in silica encapsulates, for which hydrolytic damage is also the most prominent decay factor¹⁶). Under these conditions, the simulation (Fig. 5) predicts that an error correcting code needs to compensate for a loss of ca. 30% of the unique sequences in the absence of enzymatic repair. According to the Shannon limit this requires a logical redundancy of at least 30%, and consequently, a surplus of 43% of unique sequences would have to be synthesized (surplus = $(1/(1-0.3)-1)*100$), to allow for an error-less data recovery. However, when applying a DNA repair strategy, originally lost sequences can be recovered, as shown experimentally in Figure 4c. Assuming a repair capable of recovering 50% of the sequences lost (= ca. average experimental performance of the APL mix) at these parameter choices (phys red. = 10, $t_{1/2}=2$), significantly less sequences would be missing from the pool (12%), requiring a drastically smaller logical redundancy overhead, and a lower number ($13.5\% = 1/(1-0.12)-1*100$) of surplus DNA sequences. In this scenario this leads to a significant cost reduction of ca. 200 USD / MB as DNA synthesis costs (ca. 1000 USD /MB)²⁵ is currently the main bottleneck of DNA data storage. Consequently, the application of the rather straightforward APL enzymatic repair (< 0.3 USD/reaction, Supporting Info Table 4) applied at the end of the storage horizon and directly prior to data readout, has significant impact on the synthesis cost and read out reliability of the data.